# Hydroethanolic Extract of *Urtica dioica* L. (Stinging Nettle) Leaves as Disaccharidase Inhibitor and Glucose Transport in Caco-2 Hinderer

**DOI:** 10.3390/molecules27248872

**Published:** 2022-12-14

**Authors:** Mohammad A. Altamimi, Ibrahim M. Abu-Reidah, Almothana Altamimi, Nidal Jaradat

**Affiliations:** 1Department of Nutrition and Food Technology, An-Najah National University, Nablus P.O. Box 7, Palestine; 2Department of Industrial Chemistry, Faculty of Science, Arab American University, Jenin P.O. Box 240, Palestine; 3Department of Medicine and Surgery, Università di Napoli Federico II, 80131 Naples, Italy; 4Department of Pharmacy, Faculty of Medicine and Health Sciences, An-Najah National University, Nablus P.O. Box 7, Palestine

**Keywords:** *Urtica dioica*, diabetes mellitus, medicinal plants, Caco-2, α-glucosidase, glucose transport

## Abstract

Herbal treatment for diabetes mellitus is widely used. The pharmacological activity is thought to be due to the phenolic compounds found in the plant leaves. The present study aims to investigate the phytochemical composition of *Urtica dioica* (UD) hydroethanolic extract and to screen its antidiabetic activity by disaccharidase hindering and glucose transport in Caco-2 cells. The results have shown that a total of 13 phenolic compounds in this work, viz. caffeic and coumaric acid esters (**1**, **2**, **4**–**7**, **10**), ferulic derivative (**3**), and flavonoid glycosides (**8**, **9**, **11**–**13**), were identified using HPLC-DAD-ESI/MS^2^. The most abundant phenolic compounds were **8** (rutin) followed by **6** (caffeoylquinic acid III). Less predominant compounds were **4** (caffeoylquinic acid II) and **11** (kaempferol-O-rutinoside). The UD hydroethanolic extract showed 56%, 45%, and 28% (1.0 mg/mL) inhibition level for maltase, sucrase, and lactase, respectively. On the other hand, glucose transport was 1.48 times less at 1.0 mg/mL UD extract compared with the control containing no UD extract. The results confirmed that *U. dioica* is a potential antidiabetic herb having both anti-disaccharidase and glucose transport inhibitory properties, which explained the use of UD in traditional medicine.

## 1. Introduction

Type 2 diabetes mellitus (T2DM) is one of the rapidly growing diseases worldwide that is due to lifestyle changes, urbanization, and population aging [1]. T2DM is a serious and costly disease. More importantly, the chronic complications associated with T2DM include accelerated development of cardiovascular diseases, end-stage renal disease, loss of visual acuity, and limbs amputation, which greatly affect a patient’s quality of life and contribute to the excess morbidity and mortality in individuals with T2DM [2].

While currently there is no reliable therapy for diabetes, there are numerous traditionally used herbal remedies that have long been used to help maintain hyperglycemia. *Urtica dioica* (UD) is widely used in the remedial of several diseases, in particular T2DM [3].

*Urtica dioica* L., “stinging nettle”, belongs to the family Urticaceae, a perennial plant native to Asia, Europe, America, and particularly to the Mediterranean region. *U. dioica* is known for its usage in human diet and folk medicine because of being widespread and having conjointly remarkable bioactivities [4]. The dried leaves are used to make herbal teas. Nettle leaves are eaten fresh and can be used as a potherb, soup, and herbal infusion [5]. Stinging nettle leaves represent an inexpensive and excellent source of macro- and micronutrients [6,7]. For a long time, beneficial effects of this plant have been confirmed in the treatment of many ailments including allergies, anemia, internal bleeding, wound healing, gastrointestinal tract, hemorrhage, skin, arthritis, gout, influenza, rheumatism, eczema, urinary system problems, cardiovascular system, and euglycemia [8,9].

In addition, wide spectra of biological activities of nettle leaves have been previously reported, including antimicrobial, antiviral, antioxidant, anti-inflammatory, antiulcer, hypolipidemic, and many other properties [4]. These activities were related to the presence of biologically active compounds, such as vitamins, amino acids, carotenes, fatty acids, terpenoids, fibers, and phenolic compounds [10].

T2DM management is directed toward the control of postprandial fluctuations or excursions in blood glucose, which reflect the relative rates of delivery from the intestine and disposal to the tissues [11]. Most carbohydrates are digested by salivary and pancreatic amylases and are further broken down into monosaccharides by enzymes in the brush border membrane (BBM) of enterocytes. For example, lactase-phloridzin hydrolase and sucrase-isomaltase are two disaccharidases involved in the hydrolysis of nutritionally important disaccharides. Once monosaccharides are presented to the BBM, mature enterocytes express nutrient transporters to transport the sugars into the enterocytes [12].

Alpha-glucosidase inhibitors are considered as an important treatment option in this aspect [13]. These inhibitors can be either be a product of microbial fermentation such as acarbose [14], or phytoconstituents, such as flavonoids, alkaloids, terpenoids, anthocyanins, glycosides, phenolic compounds, among others, which have been isolated from plants or algae [15].

UD was reported to have an antidiabetic effect in a clinical trial [16]. Moreover, Ziaei, et al. [17] reviewed eight randomized clinical trials and found that supplementation of T2DM patients with UD leaves significantly improved their fasting blood sugar. El Haouari and Rosado [18] have also reported beneficial properties, in addition to antidiabetic properties, such as cardiovascular improvement.

Yet, little is known about the identity and the mechanism of action of the active components of this herbal treatment. Characterizing the active components of the UD extract, as well as its cellular mechanism of action, is a vital step toward describing its role in lowering blood glucose levels in the diabetic state.

In this work, a study of the glucose-lowering in vitro effect of the UD hydroethanolic extract using Caco-2 cells has been established. In this sense, a hydroethanolic extract of UD leaves was tested for its potential activity based on disaccharidase inhibition and glucose transport in Caco-2. Besides, qualitative characterization of the phenolic compounds of UD was performed using HPLC-DAD-ESI-MS^2^.

## 2. Results and Discussion

The use of medicinal plants is a growing branch of alternative and complementary medicines used for many years to treat patients with DM. UD is a traditional plant that has long been used to lower hyperglycemia. Previous studies have provided evidence for the blood glucose-lowering effect of UD [19]. In the current study, two main issues were dealt with. First, major active compounds of UD hydroethanolic extract have been identified. Second, the antidiabetic activity of UD extract in vitro using Caco-2 cells has been estimated using three disaccharides.

### 2.1. Identification of Phenolics Using HPLC-DAD-ESI/MS^2^

The phenolics present in UD leaves extract were identified using DAD spectra and MS^2^ acceptable data. Characteristic UV chromatographic profile UD hydroethanolic extract is shown in Figure 1. In Table 1, a list of the major identified compounds is reported. The identification process was based on the data obtained from the UV maximum absorption of compounds and the MS^2^ fragmentation pattern in the negative ESI ionization mode, and by comparison with data available in the literature either regarding the bioactivity or the compounds detected in Urtica. The identified compounds were numbered according to their elution order. Although another study has shown that 45 phenolic compounds can be presented in UD aqueous methanolic extract [20], only 13 compounds were found here in reliable quantities.

Peaks **1**, **4**, and **6** presented the identical precursor ion at *m*/*z* 353 and UV spectra, with maxima at 324 nm and a shoulder at 297 nm. Thus, these compounds were assigned as caffeoylquinic acid and isomers. Compound **6** has exposed the highest peak among the isomers. This compound has been already reported in the Urtica species [20].

The DAD spectrum of peak **2** suggests a compound with a structure of caffeic acid. The compound generated a mass spectrum with a deprotonated molecular ion at *m*/*z* 311 and the MS^2^ base peak at *m*/*z* 149 after losing a caffeoyl residue (−162 Da), and secondary peaks at *m*/*z* 179 and 135, which correspond to caffeic acid and caffeic acid, lost a CO_2_ (−44 Da). Therefore, compound **2** was labeled as caffeoyl tartaric acid. This compound is reported here in Urtica for the first time.

Peak **3** had maxima at 325 and a shoulder at 293 nm and the deprotonated molecular ion of *m*/*z* 355. Based on data obtained from the UV and mass spectra, and the fragmentation pattern that showed 209 and 191 ions, which correspond to deoxyhexoside and hydroxyferulic acid, respectively, compound **3** was suggested to be hydroxyferulic acid deoxyhexoside.

Peaks **7** and **10** exhibited UV spectra profiles characteristic of p-coumaric acid structure and MS containing a precursor ion at *m*/*z* 279 and MS^2^ ions at *m*/*z* 163 (indicating coumaric acid in structure) and *m*/*z* 133 (corresponds to malic acid). Subsequently, **7** and **10** were identified as isomers of p-coumaroylmalic acid. It is worth mentioning that peak **8** has displayed a higher peak in the MS chromatogram (Figure 1).

Peak **8** (Rt 15.83 min) showed (273sh, 291sh, 351) in the UV spectrum, which is characteristic of quercetin derivative. Mass spectra showed the deprotonated molecular ion at *m*/*z* 609 and base peak (100%) at 301 (indicating quercetin). Based on the above data, peak **8** was labeled as rutin.

Compounds **11** and **13** with the deprotonated molecular ions at *m*/*z* 623 and 593 demonstrated identical neutral loss [M-H-308]-, which corresponds to rutinoside in structure (relative intensity, 100%). As a result, the fragment ions at *m*/*z* 285 (kaempferol) and *m*/*z* 315 (isorhamnetin) were detected as base peaks in the MS^2^ spectra. Thus, **11** and **13** were assigned as *O*-rutinoside of kaempferol and isorhamnetin [20].

Otherwise, the UV spectrum of compound **12** showed maxima 348 and a shoulder at 290. The MS spectrum showed the precursor ion at *m*/*z* 447 and MS^2^ product ion at *m*/*z* 285 [M-H-162]- (corresponds to Kaempferol in structure). Thus, compound **12** was identified as kaempferol-*O*-hexoside.

Isorhamnetin dihexoside malonate (Rt 17.31 min) has been suggested for compound **9**, which showed the deprotonated molecular ion at *m*/*z* 725 and the fragment ion at *m*/*z* 447 (100%) after the neutral loss of hexose moiety and a malonyl moiety (162 + 116 Da), and the fragment ion at *m*/*z* 315 (which denotes isorhamnetin in structure). This compound is being reported in UD for the first time.

The most abundant phenolic compounds presented here were **8** (rutin) followed by **6** (caffeoylquinic acid III). With fewer prominences, compounds **4** (caffeoylquinic acid II) and **11** (kaempferol-O-rutinoside) were also presented. Other phenolics were found in minute amounts.

Similar results suggested by other studies were that stinging nettles were rich in rutin, caffeoylquinic acid, and quercitrin, making it a good antioxidant, anti-inflammatory, and antidiabetic with extraordinary properties [21,22]. Recently, Pimpley et al. [23] reviewed the effect of chlorogenic acids such as caffeoylquinic acid from green coffee on obesity and diabetes. They reported many kinds of research that have confirmed the ability of these phenolic compounds to decrease patients’ fasting blood sugar, waist circumferences, BMI, and levels of triglycerides and cholesterol. Compound 12 has been already detected in different parts of the UD plant [22].

Moreover, the phytochemical profiles including caffeoylquinic acids were found to exhibit significant inhibitory activity against key digestive enzymes linked to type 2 diabetes and obesity [24].

### 2.2. α-Glucosidase Activity

UD extract at 20 µL showed inhibition of the substrate’s color change by 7% (±2%). Even though the inhibition seemed low, and more volume of the extract was needed to perform the other tests, the test has shown that UD was potentially active as an anti-glucosidase agent. The hydroethanolic solution from UD was found to have 35%, 41.5%, and 44.1% of α-glucosidase inhibition activity when using baker’s yeast, rabbit liver, and rabbit small intestine α-glucosidase, respectively [25].

### 2.3. Disaccharidase Activity

UD extract with increasing concentration showed the highest level of inhibition with maltase as 56% (1.0 mg/mL), followed by sucrase inhibition with 45% (1.0 mg/mL), then lactase inhibition with 28% (1.0 mg/mL) (Figure 2). The mean difference of inhibition level was significant between enzymes at similar UD concentrations (*p* ≤ 0.05). It is noteworthy that IC_50_ was achieved only when maltose was used as a substrate, which was between 0.2–0.5 mg/mL of the UD extract. This means that the extract contained inhibitors that compete with maltose > sucrose > lactose on the enzyme’s active sites. Moreover, higher concentrations of the extract will be needed to reach 50% inhibition when using both sucrose and lactose. Such results were consistent with other studies that showed the increasing concentration of UD extracts was associated with a higher inhibition rate of the disaccharidase [26]. However, the current study has shown a lower concentration of UD that achieved the IC_50_ than previous studies. This can be explained, partly, by the different enzymatic conditions used and, more importantly, by the fact that plants are variable in their content of active compounds. Soil, precipitation, and altitude are some of the environmental factors that play major roles in determining the plant’s chemical content. On the other hand, Caco-2 cells were reported to fully express the enzyme(s) α-glycosidases and sucrase-isomaltase, with maltose as disaccharide the highest in inducing such enzymes [27], while little is known about Caco-2 cells’ ability to express lactase [28]. Caco-2 cells are enterocytes that resemble human adult colonocytes. They express both sucrase and maltase after differentiation, which have the same sequence as human disaccharides [29].

### 2.4. Glucose Transport

Results of glucose transport are shown in Figure 3. Glucose transport into Caco-2 cells was dose dependent and affected by the UD extract’s concentration. The ability of these cells to uptake glucose was decreasing as the UD concentration was increasing in the medium. The highest inhibition was achieved with 1.0 mg/mL UD extract. This was 1.48 times less than the control, which contained no UD extract. It was reported that enterocytes are sensing the presence of glucose and absorption is mediated by GLUT2 transporter, which is moved to the apical side of the cells [11,30]. Under the influence of insulin, GLUT4 is translocated to the surface of the myocytes to mediate the glucose uptake [30]. Plant extracts including UD were found to partially exhibit their antidiabetic effects by inhibition of the GLUT2 transporting of glucose or through GLUT4 translocation [31]. The major compounds in the current study, rutin, caffeoylquinic acid, and kaempferol-O-rutinoside, were reported to inhibit GLUT2 transporter [29,32,33].

## 3. Materials and Methods

### 3.1. Chemicals

All solvents were HPLC grade and were degassed and filtrated before their application, whereas all other chemicals were of analytical reagent grade. Dulbecco’s modified eagle medium (DMEM), GlutaMAX™, was purchased from Life Technologies, Paisley, UK, while antibacterial/antimycotic solution, phosphate-buffered saline, trypsin-EDTA, glucose oxidase/peroxidase and o-dianisidine, baker’s yeast α-glycosidase, p-Nitrophenyl-α-glucopyranoside sucrase, maltase, lactase, and acarbose were obtained from Sigma-Aldrich, Pool, UK, as well as the following reagents: ethanol, hexan, formic acid, and methanol. The multi-well Tecan’s Spark® plate reader was obtained from Tecan Group Ltd., Mannedorf, Switzerland.

### 3.2. Plant Material

Stinging nettle leaves were collected from rural areas in Nablus district (the West Bank, Palestine) during the period of March–April in 2016. The *Urtica dioica* L. plant was identified at the Faculty of Sciences, Department of Biology and Biotechnology, An-Najah National University. Leaves were dried naturally in a drafted shade for a month. Dried plant material was ground using a household blender and then kept in an air-tight glass jar till usage.

### 3.3. Preparation of the Extracts

Twenty-five grams of the dried and powdered plants was soaked in a 3:1 mixture of ethanol (50%) and hexane in a well-closed Erlenmeyer flask. Then the containers were placed in the shaking incubator for 3 days at 200 rpm and the mixture temperature was kept at 25 °C. After that, the soaked materials were filtered using a semi-permeable filter, then the organic and the hydroethanolic phases were separated from each other using a separator funnel. The above procedure was repeated twice. The organic phase was eliminated, and the residue was weighed to calculate the yield. The hydroethanolic phase was placed in a rotary evaporator for one hour at 35 °C to eliminate ethanol. The hydroethanolic phase was freeze-dried and then weighed to estimate the obtained yield.

### 3.4. Phenolics Analysis by HPLC-DAD-ESI/MS^2^

Samples of UD have been analyzed by using an Agilent-1100-series high-pressure liquid chromatograph equipped with a Zorbax C18 reverse-phase column (10 mm × 4.6 mm, particle size 5 μm (Agilent Technologies, Palo Alto, CA, USA). A mobile phase (A) of acidified water (formic acid 1%, *v*/*v*) and methanol (B) was used with the following multistep gradient: 5–15% B (0–5 min), 15–25% B (5–7.5 min), 25–50% B (7.5–25 min), 50–85% B (25–33 min), and a 3 min post-run was used after each analysis. The temperature was maintained at 25 °C and the flow rate was 0.2 mL/min. The detection was done with a diode array detector (DAD) for a multi-wavelength detection detector in a wavelength range of 190–580 nm, and interfaced with an AB Sciex API-5000 MS equipped with a Turbo V ESI source (Foster City, CA, USA). The mass spectrometer system was programmed to perform a full scan in the negative ion mode. Capillary and source voltages were −10 V and 4.0 kV, respectively, and the capillary temperature was 250 °C. The sheath gas used was nitrogen at a flow rate of 8 L/min. Nitrogen was used at a normalized collision energy of 50%.

### 3.5. Cell Culture Maintenance

Caco-2 cells were obtained from the cell and culture collection at the University of Reading, UK. All cells were cultured in Dulbecco’s modified eagle medium (DMEM) with high glucose (4.5 g/mL) containing sodium pyruvate (110 mg/mL) (Life Technologies, Paisley, UK), supplemented with 10% defibrinated fetal bovine serum, 5% GlutaMAX™ (Life Technologies, Paisley, UK), and 1% antibacterial/antimycotic solution containing 10,000 units/mL of penicillin, 10,000 µg/mL of streptomycin, and 25 µg/mL of Fungizone^®^ antimycotic (Sigma-Aldrich, Pool, UK). All cells were grown in 75 cm^2^ flasks and incubated at 37 °C with 5% CO_2_ and 95% relative humidity. Media were changed every other day. After reaching 70–80% confluence, cells were split as follows: media were aspirated and cells were washed twice with 5 mL pre-warmed phosphate-buffered saline (Sigma-Aldrich, Pool, UK), then 5 mL trypsin-EDTA (0.5 g/L, Sigma-Aldrich, Pool, UK) was added and cells were re-incubated for a further 10 min. Trypsin was deactivated by adding 10 mL fresh medium followed by centrifugation at 815× *g* for 5 min. The supernatant was aspirated, and the pelleted cells were reconstituted with 1 mL fresh medium. A cell count was conducted, and 10^5^ cells/mL were recaptured in a sterile flask. Flasks were monitored daily and checked microscopically for any contamination [34].

### 3.6. Glucose-Oxidase Assay

Glucose production by disaccharidase activity of Caco-2 cells was quantified using glucose-oxidase assay [35] with some modification. Briefly, samples were diluted with deionized water to approximately 20–80 μg glucose/mL. Then 100 μL was transferred to a 96-well microplate. At zero time, the reaction was started by adding 200 μL of assay reagent (containing glucose oxidase/peroxidase and o-dianisidine, both from Sigma-Aldrich, Pool, UK) to the first tube with mixing. A 30 s interval between tubes was applied. Mixtures were incubated for 30 min at 37 °C. The reaction was stopped by adding 200 μL of 12 N H_2_SO_4_ into each well. Carefully, the mixture was pipetted up and down. The absorbance of each tube was measured against the reagent blank at 540 and 405 nm using a multi-well plate reader (Tecan Group Ltd., Mannedorf, Switzerland).

### 3.7. Measurement of α-Glucosidase Activity

Inhibition of α-glucosidase activity by the extract was measured as described elsewhere [36,37]. Briefly, 120 µL PBS (0.5 M, pH 6.8), 20 µL extract (1 mg/mL), 50 µL Backer’s yeast α-glycosidase (≥10 U/mg protein, Sigma-Aldrich, Pool, UK) solution (25 mg/mL), and 50 µL p-Nitrophenyl-α-glucopyranoside (3 mM, PNPG, Sigma-Aldrich, Pool, UK) were added to a 96-well culture plate. This mixture was allowed to stand for 40 min at 37 °C before adding 60 µL of 0.67 M Na_2_CO_3_ to stop the reaction. The absorbance value was measured at 405 nm using a multi-well plate reader with software (Tecan Group Ltd., Mannedorf, Switzerland). The α-glucosidase inhibitory activity was expressed as percentages of inhibition as:$$\mathrm{Inhibition}\;\%,=\frac{\left(Ac-As\right)}{Ac}\;100$$
where *Ac* and *As* represent the absorbance levels of the control (containing PBS, α-glucosidase and PNPG) and sample (containing PBS, extract, α-glycosidase and PNPG), respectively.

### 3.8. Disaccharidase Activity

Sucrase, maltase, and lactase activities were determined using Dahlqvist’s method [38]. Briefly, cells were seeded at a density of 10^5^ cells per well in 24-well plastic microplates. The cells were left overnight to establish adhesion followed by rinsing with 700 µL PBS. Cultured monolayers were prepared using 800 µL of substrate (28 mmol/L) and 200 µL of UD solution (final concentrations, 0.05, 0.10, 0.20, 0.50, and 1.00 mg/mL). The substrates used for sucrase, maltase, and lactase assays were sucrose, maltose, and D-(1) lactose, respectively. Acarbose (positive control) was used as a positive control. After incubation for 40 min at 37 °C, the substrate solution was decanted. The amount of glucose released from the substrates was determined using the glucose assay and measured on multi-well plate reader with software (Tecan Group Ltd., Mannedorf, Switzerland).

Decomposition of 1.0 mmol/L substrate per min was defined as an enzyme activity unit [39]. Results were expressed as IC_50_ values, i.e., the concentration of UD that decreased enzyme activity by 50%. The lower concentration that achieved IC_50_ values indicated greater inhibitory activity.

### 3.9. Glucose Transport

Transport of glucose into cells was measured at room temperature using 24-well plastic microplates seeded in advance with Caco-2 cells at a density of 10^5^ per well. The rate of glucose transport was determined by incubation of 0.4 mL UD (0.05, 0.10, 0.20, 0.50, and 1.00 mg/mL) for 10 min at 37 °C. Next, a 1.6 mL glucose solution (56 mM) was added to each well. Following incubation with UD for 40 min at 37 °C, 1 mL of solutions from each well was pipetted out and glucose concentrations were determined using a glucose assay [27,35] and measured on a multi-well plate reader with software (Tecan Group Ltd., Mannedorf, Switzerland).

### 3.10. Statistical Analysis

Data were expressed as the mean of triplicates. Assessment of difference significancy between treatment was conducted using SPSS^©^ software version 21, where a *p*-value ≤ 5% was considered as significant.

## 4. Conclusions

In this work, a total of 13 phenolic derivatives with reliable quantities were identified in the hydroethanolic extract of *Urtica dioica* L. leaves using the HPLC-DAD-ESI/MS^2^ method. The results obtained suggest that the *U. dioica* hydroethanolic extract managed to inhibit the disaccharides involved in carbohydrate digestion. Moreover, glucose transport was decreased in a dose-dependent manner. More research is still needed to separate the active components and test their bioactivity and pharmacokinetics, which will be our next step. This research contributes to the understanding of the potential of the stinging nettle use as a functional food.

## Figures and Tables

**Figure 1 molecules-27-08872-f001:**
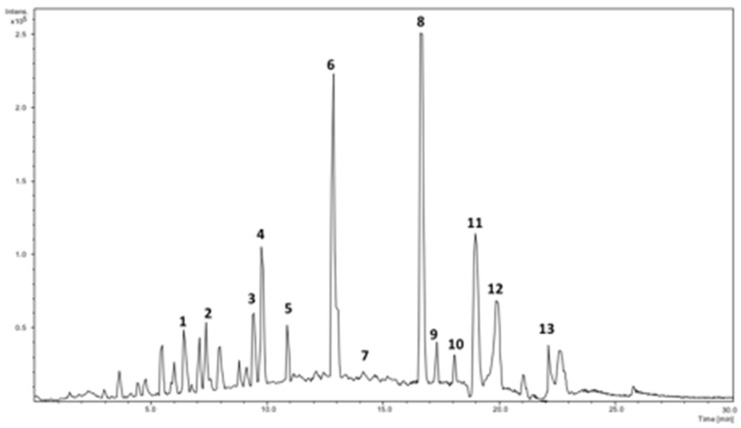
HPLC–DAD-MS profile of the stinging nettle leaves at 284 nm.

**Figure 2 molecules-27-08872-f002:**
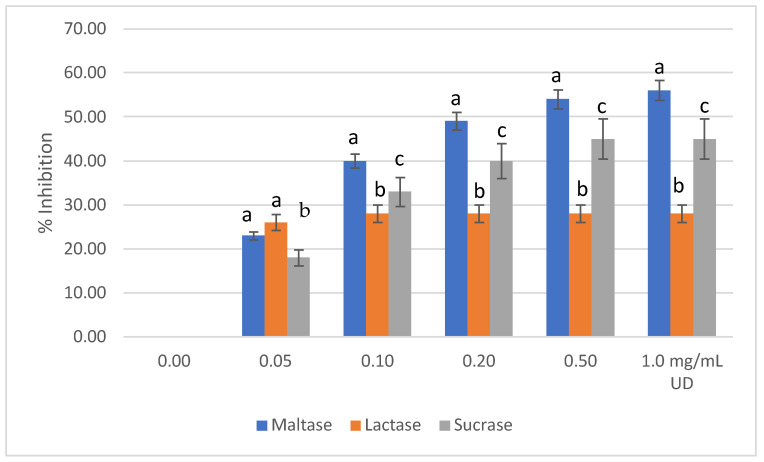
Means of inhibition % (±SD) of maltase, lactase, and sucrase activities by different concentrations of UD extract, as measured by the concentration of released glucose. Values with different letters are significantly different (*p* ≤ 0.05).

**Figure 3 molecules-27-08872-f003:**
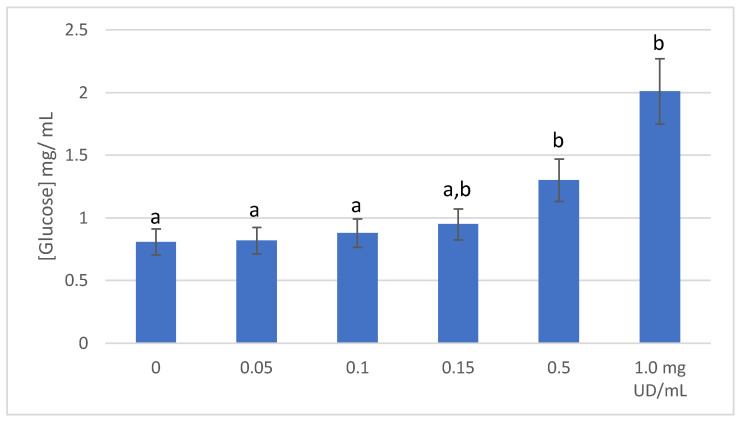
Glucose concentration (mean ± SD, mg/mL) remained in the medium after treating Caco-2 cell with different concentrations of UD extract. Values with different letters are significantly different (*p* ≤ 0.05).

**Table 1 molecules-27-08872-t001:** Phenolics characterization of *Urtica dioica* L. by HPLC–DAD–ESI-MS^2^.

Peak #	Rt (min)	[M − H]^−^	λ_max_ (nm)	Fragment Ions (% Base Peak)	Assignment
**1**	6.55	353	295sh, 325	191 (100), 179 (79), 135 (3)	Caffeoylquinic acid I
**2**	7.65	311	296sh, 329	179 (70), 149 (100), 135 (8)	Caffeoyltartaric acid
**3**	8.68	355	293sh, 325	209 (65), 191 (100)	Hydroxyferulic acid deoxyhexoside
**4**	9.85	353	298sh, 323	173 (100)	Caffeoylquinic acid II
**5**	10.89	591	291sh, 326	295 (100), 179 (30)	Caffeoylmalic acid
**6**	12.91	353	294sh, 312	191 (100)	Caffeoylquinic acid III
**7**	14.72	279	272sh, 313	163 (100), 119 (9)	Coumaroylmalic acid I
**8**	15.83	609	273sh, 291sh, 351	301 (100)	Rutin
**9**	17.31	725	266sh, 330	477 (100), 315 (25)	Isorhamnetin dihexoside malonate
**10**	18.21	279	287sh,311	163 (100), 119 (3)	Coumaroylmalic acid II
**11**	19.16	593	261, 332	285 (100)	Kaempferol-*O*-rutinoside
**12**	19.97	447	266, 290sh, 348	285 (100)	Kaempferol-*O*-hexoside
**13**	22.22	623	269sh, 296sh, 352	315 (100)	Isorhamnetin-*O*-rutinoside

## Data Availability

Not applicable.

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
