# Peer review of "Hydroethanolic Extract of Urtica dioica L. (Stinging Nettle) Leaves as Disaccharidase Inhibitor and Glucose Transport in Caco-2 Hinderer"

_molecules, 2022, doi:10.3390/molecules27248872_

Round 1
Reviewer 1 Report
Pag. 1
Introduction,
Line 37: ......diseases, in particular; T2DM [3]. ......diseases, in particular, T2DM [3].
Pag. 5
Results and Discussion
Identification of phenolics using HPLC-DAD-ESI/MS2
Line 197: .......comparison with data available in the literature. Literature ???
- Why were the molecular ion peaks for constituents 1, 4 and 6 not shown in Table 1?
Lines 217 – 220 – Line 219: Subsequently, 7 and 8 were identified as isomers of p-coumaroylmalic acid. When, where and how were identified?
- How was compound 9 identified?
Page 7
Line 261: Disaccharidase activity
Line 262: UD extract with increasing concentration showed the highest 56% (1.0 mg/mL):
- The sentence seems incomplete!
Pages 8 – 12:
The bibliography is updated and relevant, but excessively extensive for the level of the work.
The conclusions regarding Disaccharidase activity, as well as regarding Glucose transport were good, as quoted below:
Line 261:
- Level of inhibition with maltase followed by sucrase inhibition with 45% (1.0 mg/mL), then lac- 263 tase inhibition with 28% (1.0 mg/mL) (Fig. 2). It is noteworthy that IC50 was achieved only when maltose was used as a substrate which was between 0.2 -0.5 mg/mL of UD 265 extract. This means that the extract contained of inhibitors that compete with maltose > sucrose > lactose on the enzyme's active sites......
Line 284:
- Results of glucose transport are shown in Fig 3. Glucose transport into Caco-2 cells 285 was dose-dependent and affected by the UD extract’s concentration. The ability of these 286 cells to uptake glucose was decreasing as the UD concentration was increasing in the me dium............
The work in general is interesting regarding the techniques used in the determination of antidiabetic activity and the consequent results. According to the authors, In the current study, two main issues were dealt with; first, major active compounds of UD hydroethanolic extract have been identified. Second, the antidiabetic activity of UD extract in vitro using Caco-2 cells has been estimated using 3 disaccharides. However, o manuscrito it is poor in terms of identification of the main active compounds of the UD extract, as it does not show how they were characterized, just inform that they were identified by HPLC-ESI. On the other hand, it does not show the relationship of antidiabetic activity to any of the phenolic components of the extract.
Reviewer 2 Report
The manuscript entitled “Hydroethanolic extract of Urtica dioica L. (Stinging Nettle) leaves hinders disaccharidase and glucose transport in Caco-2 cells” by Mohammad A. Altamimi et al. described a study of the hydroethanolic extract of Urtica dioica L. leaves using HPLC-DAD-ESI/MS2 method and investigation for its potential activity based on disaccharidase inhibition and glucose transport in Caco-2. The manuscript may be of general interest to the researchers of this field, but the manuscript lacks some information that the author should consider and incorporate in the present form of the manuscript. Here are a few concerns that need to be addressed in the present form of the manuscript.
- The title of the manuscript should be corrected, because it now looks like a simple sentence. For example, the following correction: “Hydroethanolic extract of Urtica dioica L. (Stinging Nettle) leaves as disaccharidase inhibitor and glucose transport in Caco-2 hinderer”.
- It should be added information of commercial suppliers in “Chemicals”.
- There is only control and no reference control in the experimental studies.
- The authors should explain the choice of exactly 50 % ethanol concentration for extraction. Why not 70 %?
Reviewer 3 Report
In this work, Altamimi and colleagues aimed to address the ability of Urtiga dióica hydroethanolic extract to hinder disaccharidase activity and glucose transport in Caco-2 cells. In general, this is an interesting work, that deals with an increasingly common metabolic condition, T2DM. Despite of the whole interest of this work, there are some major aspects that should be taken into account to improve the general quality and likelihood of acceptance in Molecules.
- abstract: some numerical data should be added
- materials and methods: in the subsection related to chemicals, the complete list of all chemicals and reagents used should be added, as well as the brands and local of purchase; l. 94-95: what is the purity of ethanol and hexane?; l. 104-117: this is the first time the procedure is used? no previous studies have addressed urtiga dióica-derived extracts chemical composition? please detail; l. 119-134: please add a proper reference to support the methodology used; l. 176-183: please add a proper reference supporting the methodology used here;
- materials and methods: at the end of this section, a proper subsection of statistical analysis should be added; it is mandatory
- results and discussion: in this section, my main concerns are related to data analysis, figures 2 and 3 do not provide statistical analysis; in this sense, how can you conclude about the efficacy of the studied extract? Also, in the subsection devoted to phenolics' identification, this is the first time the chemical composition is analyzed to Urtiga dioica? please add proper data supporting the novelty of this work, and to what extent it is not the same of those already performed; l. 217 and 220: you refer compound 8; it would not be compound 10? please revise; compound 8 is rutin; l. 246: revise reference format; what other studies have been performed addressing the phenolics' composition of Urtiga dioica extracts? what are the main differences stated compared to the data obtained here? the same applied to the other experiments performed;
- conclusion: what future perspectives can be listed? what about the toxicity and pharmacokinetic data to this extract? have the pharmacokinetic aspects of this extract been already addressed?
- references should be formatted according to the journal guidelines
Round 2
Reviewer 2 Report
Dear colleagues,
many thanks for your respond to the suggestions from my side. I agree with your answers.
Wishing you all the best in future studies!
Author Response
Thanks a lot
all the best
Reviewer 3 Report
Dear Authors,
Some of the questions raised were not properly addressed and the information included in the revised version of the manuscript. In addition, the statistical analysis section should be better detailed, it is incomplete as it is.
Author Response
Dear reviewer
Thanks for your comments, could you specify the questions you would like more explanation about apart the statistical analysis. So we can upload one revised version.
Regards